# News about the Role of the Transcription Factor REST in Neurons: From Physiology to Pathology

**DOI:** 10.3390/ijms21010235

**Published:** 2019-12-29

**Authors:** Jose M. Garcia-Manteiga, Rosalba D’Alessandro, Jacopo Meldolesi

**Affiliations:** 1IRCCS San Raffaele Scientific Institute, via Olgettina 58, 20132 Milan, Italy; manteiga.garcia@hsr.it; 2Scientific Institute De Bellis, via Turi 27, 70013 Castellana Grotte, Italy; dalessandro.rosalba@libero.it; 3Department of Neuroscience, San Raffaele University, via Olgettina 58, 20132 Milan, Italy

**Keywords:** REST level, REST increase/decrease, differential gene expression, neuronal proteins, cooperation of REST with factors, miRNAs, differential role of REST in the nucleus versus cytoplasm, neurodegenerative diseases, epilepsy, brain cancers, future therapies

## Abstract

RE-1 silencing transcription factor (REST) (known also as NRSF) is a well-known transcription repressor whose strong decrease induces the distinction of neurons with respect to the other cells. Such distinction depends on the marked increased/decreased expression of specific genes, accompanied by parallel changes of the corresponding proteins. Many properties of REST had been identified in the past. Here we report those identified during the last 5 years. Among physiological discoveries are hundreds of genes governed directly/indirectly by REST, the mechanisms of its neuron/fibroblast conversions, and the cooperations with numerous distinct factors induced at the epigenetic level and essential for REST specific functions. New effects induced in neurons during brain diseases depend on the localization of REST, in the nucleus, where functions and toxicity occur, and in the cytoplasm. The effects of REST, including cell aggression or protection, are variable in neurodegenerative diseases in view of the distinct mechanisms of their pathology. Moreover, cooperations are among the mechanisms that govern the severity of brain cancers, glioblastomas, and medulloblastomas. Interestingly, the role in cancers is relevant also for therapeutic perspectives affecting the REST cooperations. In conclusion, part of the new REST knowledge in physiology and pathology appears promising for future developments in research and brain diseases.

## 1. Introduction

Intense studies, published during the last 25 years and summarized by general reviews [1,2,3], revealed that the RE-1 silencing transcription factor (REST, otherwise called NRSF) plays a critical role in the development of neurons, different from the development of other types of cells, essential for their specific phenotype and also for their functional role. In most non-neural cells, high levels of REST, accumulated in the nucleus where transcription takes place, preclude directly the expression of hundreds of genes, which remain absent during the whole cell life. In the same cells, high REST stimulates, although indirectly, the expression of many other genes. In most non-neural cells, therefore, high REST supports the expression/repression of many genes, with ensuing role in the generation of some proteins, frequent or common in these cells. 

High REST is present also in neurons and neural cells, limited, however, to their stem and early progenitor states, where regulation is peculiar [3,4,5]. For these reasons, the early cells share properties with non-neural cells. However, high levels of REST are transient in neurons as much as, during more advanced development, the level of the factor becomes much lower, mostly due to its increased turnover. Upon such reduction, neuron-specific genes undergo extensive transcription, governing the ensuing increase of specific proteins and RNAs [1,3,4], especially microRNAs (miRNAs) [5]. 

Although present in all, low REST is not homogeneous but somewhat variable in distinct types of mature neurons. Such differences contribute, directly and indirectly, to the observed variabilities of phenotype and function, observed during their life in groups and single neurons [6]. The level in neurons is not fully stable but increases transiently in some physiological states, for example, during intense stimulation [3]. In other physiological conditions, such as aging, a fraction of REST remains in the cytoplasm, accumulated within organelles such as mitochondria, where the factor is inactive [7]. In addition to physiological processes, REST governs also events dependent on processes, such as metabolic alterations, for example, obesity and insulin resistance, up to frank brain diseases, where the factor plays pathogenic roles. Moreover, in a few conditions, increases of REST do not affect but protect the survival of neurons [8]. Differentiation, therefore, is not the only site of REST action. Indeed, the factor is critical in various aspects of neuronal physiology and pathology [2,3,9,10]. 

Based on the reported considerations, knowledge of REST might have been completed a few years ago. However, the publication of new REST experimental data has continued during the last five years, important for further understanding of REST and its role in neurons. Reviews appeared recently have been mostly focused on specific aspects of REST action, concerning physiological and/or disease conditions [11,12,13,14]. So far, however, no review has recently appeared in which the presentation is comprehensive, as in our previous review [3]. The latter type of presentation is the task of the present review. We intend to illustrate the news that, the last few years have led to significant changes, innovations, and refinements in the field of REST action. These new data are presented in two sections, the first dedicated to the mechanisms and processes leading to basic discoveries, the second dealing with pathological processes including therapy. Both sections include a few subsections, dedicated to specific issues.

## 2. Physiology of REST

REST is a repressive transcription factor markedly reduced, by a considerable increase in its turnover, during the development of stem and progenitor cells. Its direct effects, taking place mostly by epigenetic mechanisms [14,15,16], involve the interaction with various modifiers, including small molecules, which affect the REST binding to chromatin remodelers as well as the ensuing nuclear reprogramming [1,2,3,17]. In addition to numerous, directly stimulated neuronal genes, low REST participates in the control of distinct regulatory factors, involved in the expression/repression of many other genes. In such indirect effects, REST often operates as a fine tuner, reinforcing or reducing various types of responses, for example, the signaling of many receptors and the secretion of clear and dense vesicles, respectively [3,18]. The neuronal low levels of REST, typical of young and mature animals, increase during aging, however, with predominant accumulation in the cytoplasm, and not within the nucleus. Physical exercise of aged patients reinforces the cytosolic accumulation of REST, accompanied by decreases in brain inflammations and increased levels of some interleukins. Moreover, under these conditions, growth factors such as BDNF tend to accumulate [19].

In addition to the concepts summarized so far, this Section covers developments and details of REST physiology characterized by recent studies. Section 1 deals with direct and indirect processes of gene expression induced by REST. Section 2 with the conversion of adult human fibroblasts into neural cells and vice-versa, previously investigated by studying processes different from the REST suppressions illustrated here. Section 3 illustrates a few forms of REST cooperation with factors analogous to the members of the Polycomb family see [3]. Cooperative factors interact with REST at various stages of neuronal life: differentiation, survival, and remodeling after injury.

### 2.1. REST Governs the Expression of Many Neuronal Genes

During the last several years, knowledge of REST in human neurons has greatly increased. In neurons, however, the number of genes identified as REST-dependent is still relatively small (around 100) [20,21]. A much larger number of genes have been identified in PC12 cells, a neuronal cell line derived from rat chromaffin cells, previously employed in almost 20,000 articles as a model of neurons. An advantage of PC12 cells has been the heterogeneity of their clones, previously characterized in detail. In several wild-type clones, the level of identified REST-dependent genes is low, largely corresponding to the genes identified in neurons. In contrast, the cells of a few other PC12 clones, defective in differentiation and secretion, are rich in the factor [18]. The population of genes expressed to similar levels in both wild-type and defective clones is of over 13,000. Of almost 2000 additional genes, half was over 10–200 fold higher in the defective compared to the wild-type clone, the other half was also differently expressed, higher, however, in the wild-type with respect to the defective clone. Interestingly, the expression of over 60% of the genes more abundant in the defective clone was found to depend directly on REST, working alone or in cooperation with other factors. In the other, over 30% abundant genes, the REST effect was indirect, mediated by transcription factors of various types. At present, the mechanisms responsible for the overexpression of genes, abundant in the low-REST, wild-type clone, have not been identified [18].

### 2.2. Neural Conversions

The main agents involved in the processes by which fibroblasts, working in vitro, are converted into neurons and vice-versa, had been identified almost 10 years ago. However, the reprogramming of fibroblasts from adult donors was not easy. In fact, the classical overexpression of the neural genes Ascl1 and Brn2, which are sufficient to reprogram stem cells, were ineffective in adult fibroblasts. In such cells, REST constitutes a barrier necessary for the production of induced neurons. Such barrier, however, is removed by microRNAs against REST. Therefore, differentiation can be achieved, in vitro and also in vivo, via the increase of two microRNAs, miR-9 and miR-124, inducing extensive reconfiguration of chromatin, with ensuing opening of regions harboring REST binding sites [22,23] (Figure 1A). In additional studies, REST and its upregulating miR-29a were found to induce the conversion from retinal progenitors to retinal neurons [24]. Overall, the suppression of REST could improve the reprogramming processes and make it suitable for biomedical applications. 

### 2.3. Cooperative Interactions

REST cooperation, critical for many functions, takes place by interaction with several regulatory factors. TRIM28, a protein of the Kruppel-associated family, is a universal co-repressor of transcription. By its cooperation with REST (Figure 1B), TRIM28 induces a decreased expression of δ catenin, a factor that promotes tumorigenesis. In neurons, however, δ catenin induces mainly the outgrowth of neurites [25]. Another cooperative factor, active during neuronal differentiation, is the C-terminal domain phosphatase 1 (SCP1), an enzyme that regulates the phosphorylation of REST. SCP1 protects REST from degradation, making possible the typical changes of its level at specific sites of differentiation (4). Plant Homeo Domain Finger protein 8 (PHF8), a chromatin regulator in many cell types, operates as a transcriptional activator. In many cases, PHF8, associated with REST at promoter regions of genes, repress or activates their expression (Figure 1C) [26]. Another neuronal REST-dependent process, the differentiation of hippocampal progenitor cells, involves inter-neuronal interactions mediated by proteins and vesicles operative by paracrine manners [27]. REST interactions with cooperative factors occur not only at early but also at more advanced stages of neuronal differentiation. Binding of the factor to the prion protein PrP^c^ is essential for the expression of subunits that integrate into the NMDA receptor (Figure 1D). This integration can ultimately induce alterations of the tridimensional structure of synapses [28]. REST participates also in axonal regeneration. Shortly after injury, the level of REST increases, participating in changes in gene expression. However, the subsequent activation of epigenetic DNA methylations rapidly results in the inhibition of REST together with its miRNA, miR-9. A consequence of the last processes is intense stimulation of axonal growth [15]. 

## 3. Role of REST in Pathology and Diseases

The role of REST and its changes in the pathogenesis of diseases such as neurodegenerative diseases (Alzheimer’s, Parkinson’s, and Huntington’s diseases), epilepsy, and schizophrenia had already been discussed in our previous review [3]. In addition, reviews, focused primarily on some of these diseases, have appeared recently [9,10,11,12] in parallel to several experimental papers. In the present section, we focus on the latter papers, in particular on the properties and consequences of their experiments. Direct and indirect expression of genes induced by REST are critical in various pathological processes. Epigenetic mechanisms and the cooperation with chromatin modifiers are also operative in REST action [14,15,16]. When the responses induced in neurons are intense, the REST effects can be dysregulated, ultimately contributing to neurological diseases. Such considerations are valid also for brain cancers. 

Another process that can be relevant for pathology, including Alzheimer’s and other neurodegenerative diseases [7,16,19], is the REST increase during aging. In such diseases affected by the factor, the levels of REST are markedly increased in the neurons of hippocampus and frontotemporal cortices. In contrast, in the neurons of dentate gyrus and cerebellum, the level of REST remains low [7]. A general panel of REST-dependent diseases is reported in Figure 2. Such Figure shows the numbers 1. of patients from diseases in which a pathological role of REST has been reported (REST-dependent diseases); 2. of all REST articles published about the indicated diseases; and 3. of the REST articles appeared during the last five years (i.e., in the 2014–2019 period). 

### 3.1. Neurodegenerative Diseases

REST, a repressive transcription factor present and active in all types of cells, plays different roles, mediated especially by depolarization and receptor activation, in neurons of specialized brain areas. Initial data about neurodegenerative diseases were reported several years ago when reviews about the mechanisms of REST and its dependent therapies were first proposed. Details about these results can be found in previous and recent reviews [3,30]. Recent developments are illustrated here.

The neurodegenerative disease intensely investigated during the last 10 years has been Huntington’s disease, caused by genetic mutations leading the insertion of long polyglutamine stretches within the sequence of the specific protein huntingtin. Such mutation affects a number of patients (in Europe 62,000, Figure 2A), much lower than the patients of other neurodegenerative diseases. An effect the Huntington’s mutation is the inhibited expression of specific genes, including the gene of the nuclear neurotrophin factor BDNF, relevant for the survival of neurons [3,11,12]. Two recent studies demonstrated that decreased REST levels in the nucleus, induced by alternative splicing of its gene [31], and decreased REST binding to the heat shock protein Hsp90 [32], reduce the accumulation of the transcription factor within the nucleus, with attenuation of its toxic effects. At present, both REST-induced events are considered as potential targets for future therapies [31,32].

Another neurodegenerative disease, in which, however, the role of REST had been considered only marginal, is Parkinson’s disease that in Europe affects 1.2 million patients, corresponding to 7.5% of the REST-dependent population (Figure 2A). Recently, two studies have been reported suggesting a possible role of the transcription factor in the pathogenesis of the disease. In the first, the Parkinson’s disease was investigated in vitro by epigenetic modulations of the REST gene expressed in the neural SH-SY5Y cell line treated with MPP^+^ [33]. In vivo the latter toxic substance is known to affect nigrostriatal neurons, thus inducing the disease. When the administration of MPP^+^ was accompanied by stimulation of REST expression by the specific drug trichostatin A, protection results were obtained, both in vitro (in SH-SY5Y cells) and in vivo (in nigrostriatal dopaminergic neurons). Therefore, the role of REST in future therapy may include possible neuroprotection [33]. In a second investigation, the role of REST was studied in aged patients, compared to controls of the same age, in which the transcription factor increases analogously, however, without parallel development of neurodegeneration [7,16]. The latter nonpathological condition, which occurs neither in Parkinson’s nor in Alzheimer’s patients, appears to depend on the localization of REST outside the nucleus. In Parkinson’s and Alzheimer’s, therefore, the development of neurodegeneration appears to depend not on major increases, but rather on the nuclear concentration of active REST [7,34]. 

The number of patients affected by Alzheimer’s disease is the highest (in Europe 7 million), corresponding to 43% of the long-term brain pathology (Figure 2A). Among the papers published recently, one has reported property of the factor relevant to the diagnosis of the disease. In a considerable number of severely affected AD patients, the level of REST, low in the blood plasma, was not increased during aging. Rather, the low plasma level decreased further with the appearance of dementia [9]. In contrast, when patients were made conscious of decreased treatments or educated to receive interventions, plasma REST level increased, and this was associated with reductions of psychiatric symptoms [35]. Low plasma REST levels can thus be interpreted as symptoms of clinical severity, while increases of these levels can be considered indicative of clinical benefit. 

### 3.2. Epilepsy

The frequency of this disease is high (Figure 2A). REST plays critical roles in epileptogenesis, mediated by the increase of its level and the ensuing repression of a subset of target genes [20]. In a severe form of mouse epilepsy, the drug-resistant mesial temporal form, REST is overexpressed and the severity of the disease is proportional to its level [36]. A subsequent study in mice investigated whether such overexpression is a consequence of neuronal metabolism increased by epilepsy. The analysis of the transcription factor was carried out in parallel to the study of cytoplasmic sirtuins, NAD-dependent class III histone deacetylases, employed to monitor the metabolic state of neurons. In mechanistic terms, epilepsy induction was largely independent of metabolic alterations [37]. Another neuronal process, known to participate in the stimulation of epilepsy, is the decrease of miR-124 [38]. When exposed to status epilepticus, the mice exhibit not only prevention of REST overexpression but also activation of microglia and of high levels of inflammatory cytokines. The role of miR-124 in epileptic neurons is not combined but rather opposed to the latter effects [38]. Another property of recently established cognitive dysfunctions of epileptic patients is due to the establishment of new alterations of hippocampal neurons governed by REST. The results confirmed a role of REST polymorphism in the modulation of cognitive functions [39].

### 3.3. Enduring Spatial Memory, Spatial Nociceptive Transmission and Hereditary Deafness

REST plays a critical role also in the pathogenesis of a few additional diseases. In male rats, enduring spatial memory problems (Figure 2A) correspond to a subset of the prolonged febrile seizures occurring in children. The mechanisms include alterations of dendrites and synapses in hippocampal CA1 neurons. Upon such pathology, blockade of REST [40] allowed treated rats to retrieve their spatial memory. The results may be important to prevent cognitive processes in individuals suffering long febrile seizures [40].

Neuropathic pain (Figure 2) involves muscarinic2 acetylcholine receptor operative on spinal nociceptive transmission. The process depends on upregulation of REST in dorsal root ganglia, with ensuing acute-to-chronic transition, essential in neurons for repression of the muscarinic2 receptor [41]. Finally, alternative regulation of REST in mechano-sensory hair cells is required for human and mouse hearing (Figure 2). Deletion of REST causes hair cell degeneration and deafness. Control of REST in the critical cells is a potential treatment for hereditary forms of the diseases [42]. 

### 3.4. Cancers

Glioblastomas and medulloblastomas, the most aggressive cancers of the brain, are not very frequent. However, they have attracted many studies and publications that have been pursued recently (Figure 2A). The role of REST in the proliferation of brain cancer cells, previously investigated in only a few cases, has expanded recently. In first such studies, the glioblastoma cancer cells, upon silencing of their REST action, stop the G1 phase of their cycle, with ensuing suppression of proliferation and migration. This was apparently due to the REST repression of a few genes (for example BBC3 and DAXX), with increased expression of other genes (for example, CCND1 and CCNE1) [43]. In a second paper, the REST inhibition by siRNAs coating of nanoparticles confirmed the inhibition of glioblastoma cell proliferation and migration [44]. Further actions of REST were found to modify the prognostic evaluations of other cancers, the gliomas [45]. Proliferation of medulloblastoma cells depends on the stimulated activity of deubiquitinase, an enzyme activator of the histone H3K9me3 methylation, critical for gene transcription. Increase of deubiquitinase was found to reduce the expression of REST target genes (examples: TUBB3, SYN1, and SCG10), with decreased proliferation of medulloblastoma cells [46]. Moreover, various drugs induced inhibition of medulloblastomas by competing for the interaction of REST with its co-repressor mSin3 [47]. In contrast, another interaction of high REST with the enzyme lysine-specific demethylase 1 (LSD1) stimulated proliferation. Inhibition of either REST or the enzyme decreased cancer aggressiveness [48]. An additional study of medulloblastoma revealing the chromatin compaction induced by REST via Akt activation identified a new potential subgroup of specific therapeutic targets [49].

Processes analogous to neural cancers occur with some endocrine cancers. In pancreatic β-cells, the level of REST is low and proliferation is limited. Progenitors are higher in REST, which remains high in diseases such as diabetes. In a fraction of pancreatic cancers, high REST represses differentiation of β-cells and thus sustains their proliferation [50]. In other neuroendocrine cells, those of hormone-refractory prostate cancer, REST regulates the androgen receptor gene that reinforces growth [51,52]. In small-cell lung cancers, REST governs variable levels of expression of many genes including the neuroendocrine gene [53]. Proliferation of non-endocrine cancer cells is slow but provides trophic support to adjacent endocrine cancer cells [50,51,52,53].

In fully non-neural cancers, cell proliferation also depends on the various levels of REST. This is the case of ovarian cancers, where the level of the transcription factor correlates to the survival of patients [54]. In epithelial cells, REST level is high. Its loss is bound to cancer, its recurrence, and poor prognosis [54,55]. Finally, in liver cancer cells, REST level is variable. In all cases, however, its main localization is in the cytoplasm, relevant for liver cancer cell proliferation. In this case, the oncogenic effect of cytoplasmic REST is paradox and not yet explained [56].

### 3.5. Therapy

During the last few years, the therapeutic potential of REST has attracted attention concerning both specific conditions and their deacetylases (examples in [57,58]). Recent therapeutic attempts have been included in numerous studies reporting neurodegenerative diseases [31,32,33,34] and epilepsy [38,39]. These studies, however, concerned primarily pathology, not therapies of diseases.

The above conclusion is inappropriate for some cancer studies. In the medulloblastomas of Reference 47, the REST-mSin3 cooperation, important for the effects of the factor, was prevented by drugs that, however, are still under investigation [43,48,59]. Among the drugs investigated, three antidepressants and antipsychotics exhibited clear chemo-type activity, suggesting the possibility of new therapies based on dysregulation of REST function [46]. Additional drugs, active against various cancers, were inhibitors of the epigenetic function of REST, affecting the viability of medulloblastoma cells [47]. Other anticancer drugs operate by binding critical enzymes, LSD1 [48] and deubiquitinase USP37. Blockers of an epigenetic complex including USP37 and REST are considered as anti-medulloblastoma factors [46]. Further suggestions of anticancer therapy emerged from drugs active against another brain cancer, glioblastoma. Its inhibition by block of REST, gene transcription was of potential relevance for future therapies [43].

Additional results were of interest to the diagnostic point of view. A few drug inhibitors were shown to affect various REST complexes [58]. Another therapeutic approach, based on the REST demethylation, was active in the generation of a factor active against breast cancer [60]. Eye studies about REST were carried out by investigating the proliferation and differentiation of progenitor cells already employed for retinal degenerative diseases [24]. Two final articles referred to the brain action of lithium (Li). Its effect is important, relevant for neuronal survival upon stroke. The mechanism of this effect, however, was not clear. Recently, a study showed that brain protection depends on nuclear REST degradation, upon which Li promotes post-ischemic neuroplasticity and angiogenesis [61]. The other study reported that known protection of synaptic damage by Li is due to its binding to nuclear REST in competition to a specific prion protein. Li, therefore, operates as a therapeutic agent active against prion-based diseases induced by REST degradation [62].

## 4. Conclusions

This review shows that, during the last few years, REST has still attracted considerable interest, concerning its physiological and pathological properties. Among physiological properties, we have emphasized various new findings, concerning a large number of REST-dependent genes [18]; the REST-dependence of the conversion from epithelial to nerve cells and vice-versa; the cooperation of REST with multiple active agents related to its epigenetic activity. Another property of interest, i.e., the REST alternative splicing, has not been discussed in general terms [63]. Here, however, it has been considered in several specific cases [31,42,45,58].

The physiological aspects of REST action, included here and in our previous review [3], cannot be considered independent from pathology. In fact, knowledge in the first field has been relevant for the interpretation of several important neurodegeneration diseases, epilepsy, and cancer diseases. Additional REST-dependent effects, induced by brain injuries, affect memory, pain, and deaf. In various pathological conditions, the effects of REST were variable, contributing in some cases to cell restoration, in other cases to the severity of lesions. The final section of this review, dealing with therapies, was short, concerning brain cancers, such as those treated with already known drugs or exposed to REST associated with various complexes.

Many experimental results published in the last five years, summarized in the present review, have increased significantly our knowledge of REST. However, such knowledge is still incomplete. In our opinion, therefore, studies of the factor will continue during the next several years. For these developments, many present results will be useful, especially for the identification of the mechanisms and molecular complexes sustaining the activity of REST. Moreover, the integration of the present physiology and pathology will be relevant for the progress of therapy, the important area least developed during the last few years. Future results about REST will be interesting. In other words, they will be essential for future research, especially for the identification of new factors critical for the diagnosis and therapy of brain diseases.

## Figures and Tables

**Figure 1 ijms-21-00235-f001:**
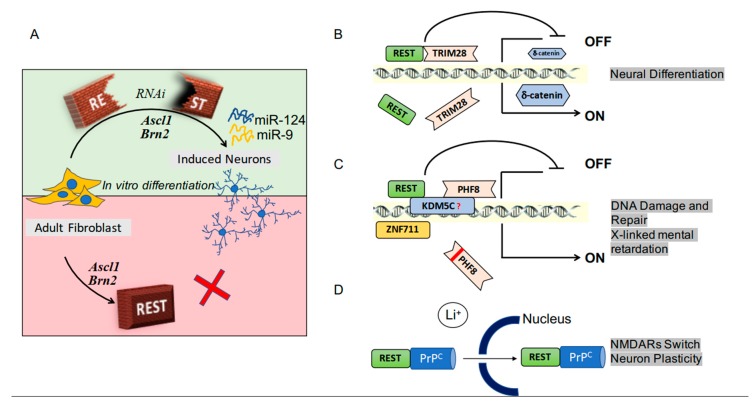
RE-1 silencing transcription factor (REST) activity in neuronal physiology. Panel (**A**) illustrates the role of REST in the conversion of fibroblasts into neurons. Overexpression of the genes Ascl1 and Brn2 favors the process, which is blocked by REST (see the bottom of (**A**)). The latter effect of REST, however, is prevented by specific RNA inhibitor (RNAi, see the top of (**A**)). The ensuing activation of two miRNAs, miR-9 and miRNA-24, act by inducing a reconfiguration of DNA, necessary for the opening of the appropriate REST binding sites. Panels (**B**–**D**) show examples of REST cooperation with regulatory factors. In (**B**) the participating factor is TRIM28, a universal co-repressor of transcription. Its direct binding to REST reduces the expression of the δ catenin, a protein that in neurons, at variance with the other cell types, induces differentiation and activation of various functions. Panel (**C**) shows that the REST-PHF8 interaction, integrated into a complex, binds the promoter regions of genes, inducing or blocking their expression, with ensuing either repair of DNA or brain defects. Finally, the cooperation of REST with the prion protein PrP^c^ stimulates their traffic to the nucleus, thus regulating the expression of NMDA receptor subunits, relevant for neuronal plasticity and the tri-dimensional organization of synapses.

**Figure 2 ijms-21-00235-f002:**
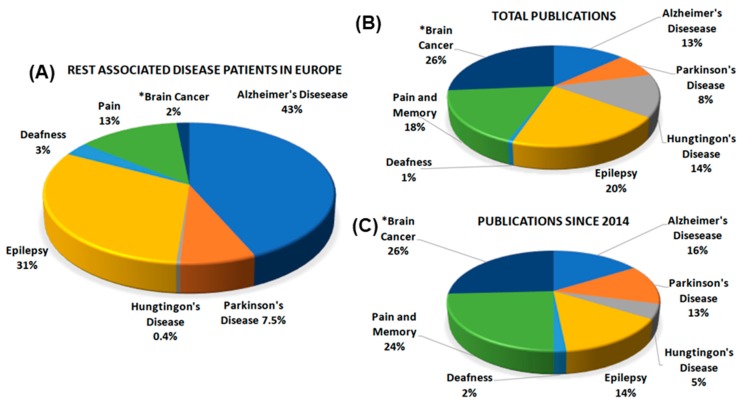
Pie Charts illustrating the patients of several brain diseases and their publications. Chart (**A**) illustrates the expression and the distribution in Europe of the patients affected by brain diseases that previous studies have reported affected by REST (REST-associated diseases). The patients of the various diseases are presented here as fractions of the 100% of Pie Chart. Among neurodegenerative diseases, Alzheimer’s accounts for 43% (=7 million), Parkinson’s for 7.8% (=1.2 million), Huntington’s for 0.4% (=62,000) of total patients (Data from the EBC Consensus Documents on European Brain Research). The additional slices illustrate the expression of other REST-associated diseases: epilepsy (31% = 5 millions); pain and memory (13% = 2 millions); and genetic deafness (3% = 0.4 millions) (Data from Epilepsy out of the Shadows). Brain cancers, accounting for 2% of the patients, have been estimated from the data of the National Cancer Institute of the USA (https://seer.cancer.gov/statfacts/html/brain.html), recalculated for the European population [29]. Pie Chart (**B**) illustrates the total publications of the same diseases, expressed in terms of percentages as in Chart (**A**); Pie Chart (**C**) to the publication of the same diseases appeared during the last 5 years. Notice the differences between the publication data of Charts (**B**,**C**) with respect to the patient data of Chart (**A**). Major such differences are those of cancers (2% patients accounting for over 25% of the publications in both (**B**,**C**)) and Huntington’s (from 0.4% to the 14 and 5% of publications). On the other hand, 43% Alzheimer’s patients correspond to only 13–16% of publications. Distinct differences, present between total and recent publications (charts **B**,**C**), are visible for Parkinson’s and deafness that increase from 8% to 13% and from 1% to 2%. In contrast, the B to C changes of Huntington’s and epilepsy do not increase but decrease: from 14% to 5% and from 20% to 14%, respectively.

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
