# Peer review of "News about the Role of the Transcription Factor REST in Neurons: From Physiology to Pathology"

_ijms, 2019, doi:10.3390/ijms21010235_

Round 1
Reviewer 1 Report
The authors present a review article entitled: "News about the Role of the Transcription Factor REST in Neurons: from Physiology to Pathology".
First a disclaimer: I am not working in the field of REST which is probably why I am a perfect reviewer for a review article. The purpose of a review article is to give a concise overview about the field of interest and guide the readers’ attention to crucial articles in the field.
The first chapter (Introduction) seems very general. The authors emphasize that they will only highlight recent advances.
A general theme throughout the presented review article is that the authors describe a function or an “important” role of REST in a cell type and mention that REST causes “various responses”. Here are some examples:
Row 99: “…REST often operates as a fine tuner, reinforcing or reducing various types of responses.” – which responses? Please specify…
Row 166: “… the neuronal levels of REST are variable: sometimes they are unchanged, sometimes they are increased or even decreased.” – then please specify in which case it is increased/decreased/unchanged.
Row 221: “…cognitive dysfunctions of epileptic patients was due to genetic alterations governed by REST.” – which genetic alterations?
Row 240: “This was apparently due to the REST repression of a few genes, with increased expression of others.” – which genes?
Row 246: “… was found to reduce the expression of REST target genes.” – what are REST target genes? Name some examples. How many are there.
Row 258: “…, REST regulates critical properties.” – if these properties are so critical, why not mention them?
Row 258-259: “In small lung cancers, REST governs variable levels of gene expressions.” – I am not from the cancer field either, but I believe that is supposed to mean: “small cell lung cancer”. AND: which types of genes?
Row 265: “… relevant for liver cancer cell proliferation.” – How does that work?
Row 273: “ … was prevented by already available drugs.” – Which ones?
Row 297: “… a large number of REST-dependent genes.” – the authors never mention which genes are dependent on REST.
It seems like the authors list publications from the previous years and just mention the topic of the paper or the title but no functional aspect or result is ever discussed.
Another problem is that the authors often cite review articles (e.g. row 180). It is not very inventive to cite secondary literature in a review – one should always refer to primary literature.
The authors also frequently refer to their previous review article (Ref. 3, e.g. in row 96, or 180). If I am new to the field, I want to read a concise review – I don’t want to be referred to another review where, maybe, I will find the information I am looking for.
In section 2.1, the authors write about “REST governs the expression of many neuronal genes”. First (as mentioned before) “many” is as specific as it gets – please specify. Second: this section is about one single publication in PC-12 cells – that is a cell line which can be differentiated into something that looks like a neuron – but these cells aren’t actual neurons.
Section 2.2 deals with reprogramming fibroblasts into neurons. The authors should emphasize that they report purely in vitro data. And they should clarify what that section has to do with physiology.
The introduction to Section 3.1 reads: “… REST a repressive transcription factor present an active in all cell types…” – I have read this several times before, rather than repeating that sentence, could the authors mention a list of genes that are regulated?
In row 189 the authors describe something in SH-SY5Y cells and immediately mention in vivo data. Again, SH cells are a cell line – not actual neurons – the authors may want to clarify this section and describe the results of the cited publication a bit better.
In row 218: the authors write: “…mice exhibit not only a prevention of REST overexpression, but…” – I am sorry, but this sounds like I should expect such an overexpression. Why should I?
In row 229: The authors mention neuropathic pain and “muscarinic2 receptors”. Can the authors maybe explain the link between M2 receptors and neuropathic pain? And may I refer the authors the IUPHAR database (https://www.guidetopharmacology.org/GRAC/FamilyDisplayForward?familyId=2): the official name is M2 muscarinic acetylcholine receptor.
In row 287: the authors mention the role of lithium and their importance in stroke. – I thought lithium is used as a treatment for bipolar disorder – I have never heard that it should be important in stroke (it is definitely not a first-line therapy). Lateron the authors mention prion-based diseases, but the section is introduced by “stroke” – It reads as if stroke were a prion-based disease. The authors should clarify that and, maybe, mention a few prion-based diseases (especially those where REST is involved).
Figure 1 deals with 3 very specific examples. The authors should describe the figure legend a bit better. A more general figure would be interesting. May be the authors can emphasize why they picked these three examples.
The data from figure 2 would work better in a table (if at all) – the percentages are anyways mentioned in the text.
Overall, I find this review not very informative. As a new reader to the field I would have to continue searching for other articles to get information. In addition, I cannot find out from this very review which other articles might be interesting to read.
Author Response
The reviewer, in her/his opinion a non-expert of REST, criticized repeatedly our review as incomplete, missing many important properties of its target. We do not agree. In our work, we had followed for over 10 years the development of knowledge about the factor. Therefore, for us the illustration in neurons of many aspects of REST would have not been difficult. However the properties known before 2015 had already been reported in reviews by us and others. Our task, therefore, was not to illustrate the whole or at least many properties of the factor but offer neuroscientists the increases of REST knowledge occurred during the last 5 years. Such task was emphasized in the title, the introduction and the conclusion of the present review. The other suggestion of the reviewer, the completion or explanation of many sites of the review has been largely accepted with the introduction of the requested changes. Specifically, changes were made following the suggestions and criticisms raised by the reviewer in the following sites:
lines 99, 166, 218, 221, 229, 240, 246, 258, 258-259, 265, 273, 297.
In addition, we asked an American friend, a colleague of the Institute, to read the paper, and we introduced the changes suggested by her/him. A small number of mistakes present in the first text have also been corrected.
Reviewer 2 Report
This review is timely as many questions remain unanswered about REST functions in neurons under physiological and pathological conditions.
The present review consists of a series of citations and excerpts, gathering a lot of information. However, the main task with a scientific literature review is to organize and summarize the references in such a way that they reveal the current state of knowledge on the chosen topic and, in the context of a new study, establish a systematic basis for the research. The review should point out both congruent points and contradictions, as well as explain inconsistencies, e.g. different conceptualizations or methods. Studies that are of particular importance to research should be described in detail, although those that yield comparable results can often be grouped and briefly summarised.
The conclusion of the present review appears to be a very simple summary of the review and the last paragraph says nothing about anything. The conclusion should be a concluding section, in which the consequences of the review, proposals for new hypotheses and concrete lines of research for the future are presented.
Authors should rewrite their review following the above suggestions.
In addition, authors are encouraged to introduce a figure(s) with a supposed regulation of REST in Huntington’s, Parkinson’s and Alzheimer’s disease. Moreover, as inhibitors or activators of REST might open the door for development of novel therapeutic strategies to ameliorate the neuronal death and impaired cognition associated with neurodegenerative disorders, a mechanism of those inhibitors/activators could be proposed in those diseases from the revised articles presented and the discussed and argued synthesis of the results.
Author Response
Reviewer 2 found our review of some potential interest. In his/her opinion, however, the presentation is entirely inappropriate because of its limitations and also because of its strategy. This because "the task of scientific reviews is to organize and summarize the references in order to establish a systematic basis of the research". In these reviews "important studies should be described in detail". In contrast, in our review "the conclusion appears to be a very simple summary". In conclusion, therefore,"the authors should rewrite the review following our suggestions".
We agree that our review was not developed by the general strategy described by the reviewer. In addition, we do not exclude that among recent reviews by others, some have been developed according to such strategy. However, such strategy is not a rule. In other words, our strategy is not forbidden. Rather, excellent reviews by other authors and laboratories, written by a strategy analogous to our, have been of great success in the scientific community. Among the previous reviews by Meldolesi and co-workers many have obtained hundreds, in 3 cases over 1000 quotations. We are convinced that also our present review, focused on the recent developments of REST knowledge, will be well received.
Finally, the reviewer found our review badly written in English language and style. We have already commented on this criticism in our answer to Reviewer 1. In a few words, in view of our long experience in the field we do not agree with this criticism.
Reviewer 3 Report
The review is well constructed and providing a comprehensive summary about the REST gene and its functional role in neuronal disease. It covers informative studies related to this factor. There are few minor concerns for further improvements.
-Regarding to Figure 2, since the data is from EBC consensus documents, authors need to clarify how they define REST associated or not, what the standard to count? Also, for the publications’ report, via key words searching or other approaches?
Author Response
We are grateful to the reviewer for his/her positive evaluation of our English and style and for the general opinion about our review, which is "well constructed, providing a comprehensive summary about the REST genes and its functional role in neural diseases. We have also re-analyzed the presentation of Fig. 2, in particular the mechanisms employed and the values obtained. This is now corrected in the Fig. 2. Moreover we have corrected the few spell mistakes found in the text.
Reviewer 4 Report
The present review discusses role of the transcription factor REST in physiology and pathology. It delivers a good overview about the present status of REST function in these two areas. The manuscript would profit from a couple of considerations and some rephrasing.
Specific points:
On lines 65f. it is stated that REST levels are controlled by metabolism. Two points here: i) 'metabolism' means 'turnover rate'? Please specifiy. ii) If known, it would be interesting to briefly add how the metabolic stability of REST is regulated. On line 74 it is mentioned that REST accumulates in 'organelles'. If it is known which, it would be good to mention this. Chapter 2.1 deals with the discovery of REST function on gene expression in PC12 cells. It would be good to compare these classical findings with newer data in neurons. Specifically, are all neuron types affected by such REST regulation, how many genes are regulated? And: how is REST regulated itself? Chapter 2.2 talks about the function of REST in determining neural fate. It would be good to add here the developmental perspective: when does regulation of REST occur during ontogenesis, is REST involved in determining neuron type? On line 201, AD is called a 'REST-deopendent pathology'. How strong is the evidence for this? It is clear that there is some alteration in REST levels in AD patients, but would modulation of REST in neurons cure AD? Please specify or modify wording. On a few occasions, e.g. line 226, 'blockade of REST' is mentioned. How does that occur? Please add this information. Finally, according to the manuscript, REST plays a pivotal role in a plethora of diseases, including some endocrine cancers. The authors should help the reader in solving the conundrum of a protein that initially appears to be so specific to determining neuronal fate and then is found to have such a wide-spread role. Please comment.Suggested rephrasing:
The sentence on lines 60f. is redundant, same for line 90f. The statements on lines 102-104 need rephrasing for clarity. The following paragraph should be shortened considerably, it does not add any critical information. The sentence on line 113 is ambiguous and needs to be rephrased. line 184: ...reported in are... there is a syntax error. line 205: ...mindfulness-based reductions... reduction of what? line 224: ((Fig. 2A) correct parenthesis line 244: Additional cancer cells investigated by REST... needs rephrasing line 315f.: Statements like 'Future results about REST will be innovative.' are pure speculation and don't read well. line 329: Error in citation 3, wrong title.Author Response
We are grateful to the reviewer for his/her accurate and detailed work made on our review. The general evaluation as positive starting with two general overviews concerning the physiology and the pathology. The English of the presentation is evaluated positive, as clear from the stars of evaluation. A number of critical points, raised in various areas, have been corrected in the text.
Line 65f: Here the metabolism deals with changes of neuronal metabolism that can be induced in various patients. This is now specified. Line 74: when concentrated in the cytoplasm REST accumulates in various organelles, including mitochondria. Chapter 2. The mechanism of REST down-regulation in neurons is reported in the first and second lines. 2.1 The mechanisms of REST-dependent gene expression in neurons are now specified as requested. 2.2 REST is low and thus operative in all neurons, however it does not play a role in their differentiation. 201. Now the REST dependence of diseases is explained in Figure 2. In the text the section has been corrected. 226. The block of REST can be made by the use of drugs active also on other targets. This can be positive in various experiments, however it is not yet for therapy, as discussed in the section Therapy.
Suggested rephrases: we are grateful for the indications where corrections were made, specifically at 60f, 90f,113,184, 205, 315f, 329, citation 3, wrong title. All these, and also a few others chosen by us, have been corrected.
Round 2
Reviewer 1 Report
The authors present a revised version of their review article entitled “News about the Role of the Transcription Factor REST in Neurons: from Physiology to Pathology”.
The revised version is much clearer. Now chapter 1 (Introduction) presents a concise summary of current REST knowledge (maybe it is just rephrasing that makes it clearer). This way it is also suitable for new readers to the field as it helps them to understand the topic.
Chapter 2 “Physiology of REST: Mechanisms and Processes” greatly benefited from the contribution of the American colleague. It nicely summarizes the recent developments. In this regard, it would be interesting to have a table of genes or processes that are regulated by REST.
Chapters 3 (Role of REST in Pathology and Diseases) and 5 (Conclusion) is still a little bit confusing and would benefit from editing by the American colleague.
Few other remarks:
There is no chapter 4.
And I wish to reply to the authors response: The authors emphasize that they only wish to highlight recent advances – the authors have highlighted this intend plenty of times in the first version of their review. The authors seem to propose that I have not understood the scope of the review (advances of the previous 5 years)… I have read and understood the scope of the review, however, I think that it is necessary to give at least a small introduction into the previously known knowledge.
As I have mentioned before: the current version is much clearer in this regard.
Author Response
We are grateful to the reviewer for his/her positive comments to the revised version of our review.
Reviewer 2 Report
The manuscript has been sufficiently improved in order to be published in IJMS.
Author Response
We were pleased to read that the reviewer found our revised version significantly improved with respect to the original version.
Reviewer 4 Report
My concerns have been addressed adequately.
Author Response
We are grateful to reviewer 4 for his/her criticisms to our initial version, with numerous comments that we had accepted, with the great improvements observed in the present version.